# *Histophilus somni* as a Unique Causative Agent of Puerperal Metritis (PM) in a Third-Lactation Holstein Cow

**DOI:** 10.3390/vetsci11030117

**Published:** 2024-03-05

**Authors:** Jéssica Molín, Andrea Ainoza, Ramon Armengol

**Affiliations:** 1Department of Animal Science, ETSEAFIV, University of Lleida, 25198 Lleida, Spain; jessica.molin@udl.cat; 2Department of Veterinary Diagnosis, Eurofins Convet S.L.U., 25005 Lleida, Spain; andrea.ainoza@convet.net; 3Agrotecnio Research Center, ETSEAFIV, University of Lleida, 25198 Lleida, Spain

**Keywords:** cattle, uterine infections, *Histophilus somni*, metritis

## Abstract

**Simple Summary:**

This paper reports a case of puerperal metritis (PM) caused by *Histophillus somni* (*H. somni*) in a Holstein cow. PM is characterized by an abnormally enlarged uterus and a fetid watery red-brown uterine discharge within the first 21 days postpartum, showing clinical signs associated with systemic illness, such as decreased milk yield, dullness, toxemia, and fever (>39.5 °C). Cows with PM need fast and complete treatment to prevent very serious consequences for the cow’s health, which could lead to death. A pure culture of *H. somni* was obtained and identified in the uterine discharge. To the authors’ knowledge, this is the first report of *H. somni*’s ability to act as a unique causative agent of puerperal metritis, suggesting under-reporting or lack of diagnosis.

**Abstract:**

This manuscript aims to report the clinical and laboratory diagnosis of puerperal metritis (PM) in a dairy cow, caused by *H. somni* as a unique pathogen. The cow showed signs of systemic illness, including a sudden drop in milk production, a rectal temperature of 40.4 °C, tachypnea, dehydration, and completely fluid, brown, and fetid uterine discharge. Pure cultures of *H. somni* were identified and submitted to the Kirby–Bauer disc diffusion method for antibiotic sensitivity. The study showed that *H. somni* was resistant to tetracyclines and cephalosporins (Ceftiofur), antibiotics commonly used to treat uterine infections in dairy cows. To the authors’ knowledge, this case describes for the first time PM caused by *H. somni* as a primary pathogen. Our results should lead to the inclusion of *H. somni* as a primary pathogen of metritis in laboratory diagnoses on a routine basis, which, in turn, may help to elucidate the incidence of *H. somni* as a causative agent of uterine infections in cows. If the incidence of *H. somni* is remarkably high or frequent, researchers could consider the use of commercial vaccines nowadays destined for the prevention of bovine respiratory disease and which could perhaps be effective in the prevention of reproductive pathology caused by *H. somni*.

## 1. Introduction

Immediately after calving, bacterial contamination of the uterine lumen is very common [1]. This fact, combined with local inflammatory phenomena and the cow’s low immune response, means that the possibility of suffering some type of uterine pathology during the first 60 days after parturition is very high [2].

One of the most common uterine diseases is metritis, pathologically defined as the inflammation of all layers of the uterine wall, showing edema, congestion, infiltration by leukocytes, and myometrial degeneration [3]. Clinically, two types of metritis can be differentiated: puerperal metritis (PM) and clinical metritis (CM), both of which are characterized by an abnormally enlarged uterus and a fetid watery red-brown uterine discharge within the first 21 days postpartum. However, cows with puerperal metritis show clinical signs associated with systemic illness, including decreased milk yield, dullness, toxemia, and fever (>39.5 °C), and require fast and complete treatment to prevent serious consequences for the animal’s health, occasionally leading to death [3]. This treatment is usually based on antibiotics and anti-inflammatory and support therapy. Multiple factors may trigger metritis in cattle, such as the calving of twins, dystocia, c-section, retained placenta, hypocalcemia, or ketosis. Furthermore, bacteria may enter the uterus through skin regions, the environment, and iatrogenic contamination can even occur. The bacteria typically associated with uterine diseases are *Escherichia coli*, *Trueperella pyogenes*, *Fusobacterium necrophorum*, *Prevotella* spp., and *Bacteroides* spp. [4].

*Histophillus somni (H. somni)* is a Gram-negative coccobacillus that has been isolated in the respiratory, nervous, and reproductive systems of cattle [5,6]. The most frequent clinical expressions are respiratory and neurological disease (including undifferentiated fever, fibrinosuppurative bronchopneumonia, diffuse pleuritis, and thromboembolic meningoencephalitis, followed in some cases by acute myocarditis, polyarthritis/tenosynovitis, abortion with placentitis and fetal septicemia, epididymitis-orchitis, and ocular infections [6]. Furthermore, it has been described as a commensal and pathogenic organism, and it has also been described as a causative agent involved in infertility and endometritis in cows [7,8]. Briefly, *H. somni* not only inhibits phagocyte function, but is also cytotoxic for macrophages, and includes other evasive mechanisms such as decreased immune stimulation and antigenic variation [9].

To the authors’ knowledge, *H. somni* has never been reported as a unique causative agent of PM in a cow. Therefore, this report describes the clinical and laboratorial diagnosis and treatment of PM in a dairy cow caused by *H. somni*, which might be a potential pathogen underestimated on farms. Our findings suggest that *H. somni* should be routinely subjected to laboratory diagnosis of PM in cows and open the door to investigate whether vaccination strategies against *H. somni* for bovine respiratory disease could be effective to avoid serious uterine infections caused by this microorganism.

## 2. Case Presentation

A 5-year-old Holstein cow was the case subject. The cow was from a high-production commercial dairy farm located in Catalonia (northeast Spain). The farm was a closed farm, fenced and with restricted and controlled movements of vehicles and persons. The farm was free of BVD and under a strict vaccination program against IBR with a live marked vaccine. The farm milked an average of 700 lactating Holstein cows with an average production of 11,400 kg of milk (3.6% Fat and 3.3% Protein) in 305 days by cows. Cows were housed in five free-stall barns with compost and sawdust bedding: one for high-production multiparous cows, one for low-production multiparous cows, one for first-lactation cows, one for cows between 0 and 21 days in milk, and one for sick cows. All pens guaranteed ≥12m^2^ of dry bedding/cow, ≥10 cm of drinker/cow, different drinking points, and enough feeding space to avoid competition between cows. Cows were fed an ad libitum total mixed ration consisting of corn silage, grass silage, and concentrate. Cows were milked three times per day in a robotic rotary milking parlor (GEA, Düsseldorf, Germany). All management, clinical examinations, treatments, and artificial inseminations were carried out using a sorting door at the exit of the milking parlor in a specific pen with head lockers designed for these tasks. Cows were never head-locked for more than 20 min.

All cows underwent a general clinical examination carried out by a veterinarian (Lleidavet SLP, Lleida, Spain), including rectal temperature and a review of the data provided by the farm software (Dairy Plan C21 Version 5.3, GEA, Düsseldorf, Germany) every 24 h, rectal palpation, and observation of uterine discharge every Tuesday and Friday between 1 and 30 days postpartum. Uterine discharge was observed through rectal palpation massage to minimize not only contamination of the vagina and uterus but also microtrauma in the area. Quantity, color, proportion of pus, consistency, and smell were evaluated [3]. Quarterly, samples of uterine contents were taken from cows suffering from uterine disease through uterine lavage and subjected to laboratory diagnosis to identify the pathogens involved and monitor antimicrobial sensitivities. The most common pathogens obtained in past analyses on this farm were *Eschericia coli*, *Proteus* spp., and *Trueperella pyogenes*. *H. somni* was never isolated from uterine or vaginal content on this farm before. This farm reports an average of 8% of metritis for the last 5 years (years 2019–2023), calculated as (cases of metritis/number of calvings) × 100.

This farm is endemic of *H. somni*, since it has been commonly diagnosed on this farm for the last 10 years in routine sampling for BRD in calves and heifers (deep nasal swaps and bronchoalveolar lavage), in combination with *Mannheimia haemolytica*, *Pasteurella multocida,* and *Mycoplasma bovis*. Furthermore, sporadic cases of neurological disease, compatible with histophilosis, have also been clinically diagnosed in youngstock.

The cow was initially diagnosed with PM at day 12 postpartum, when the farm software (Dairy Plan C21 Version 5.3, GEA, Düsseldorf, Germany) alerted to illness suspicion for this cow based on a >40% milk drop for the last 24 h. Clinical examination showed a rectal temperature of 40.4 °C, tachypnea without evidence of respiratory disease at auscultation, moderate dehydration, and uterine discharge, which was completely fluid, brown, and fetid. The cow did not suffer from diarrhea, a displaced abomasum, or ketosis (β-hydroxybutyrate -BHB- concentration in blood = 0.5 mmol/L) (FreeStyle Optium Neo, Abbot Lab, Abbot Park, IL, USA). Subclinical hypocalcemia was not monitored on the farm. The veterinarian took advantage of this case of PM to send routine samples to the laboratory for bacteriological control and antimicrobial sensitivity in metritis cases on the farm. Briefly, samples of uterine content were collected through a sterile catheter with a sterile 25 mL syringe, after cleaning the perineal area with water and neutral pH soap and disinfecting it with alcohol at 70°. Samples were taken directly from the uterus after passing through the uterine cervix.

After sample collection, the cow was treated as protocolized on the farm: oral calcium (Bovicalk, Boehringer Ingelheim, Germany) every 24 h for 2 days in order to treat or prevent hypotetic hypocalcemia and help uterine involution, plus one injection of 37.5 mg of sodium selenium and 1250 mg of α-Tocoferol acetate (Hipravit Selenio, HIPRA, Amer Spain); a non-steroidal anti-inflammatory treatment of meloxicam 0.5mg/Kg of body weight (BW) via a subcutaneous route (Metacam 40 mg/mL, Boehringer Ingelheim, Ingelheim/Rheim, Germany); and an antimicrobial treatment of 15 mg/kg of BW of amoxycillin trihidrate every 48 h by intramuscular route three times (Amoxoil Retard, SYVA, León, Spain). Injections were applied on the neck and the volume of injection was always lower than 20 mL on the same inoculation point. This specific antibiotic was chosen considering previous results of metritis from routine analysis carried out on the farm.

A sample of uterine content was immediately submitted to the laboratory (CONVET, Lleida, Spain) in a 120 mL sterile container under refrigerated conditions (4–8 °C) (Figure 1 and Appendix A). The sample was subjected to a microbiological analysis based on plate culture with blood agar and chocolate agar (Biomérieux Ref. 43101, Marcy-l’Etoile, France) for general pathogen identification. These media support the growth of all the bacteria that commonly cause or participate in metritis in cattle [10,11]. After the isolation of the microorganism grown on the plate (Figure 2), a particular identification of *H. somni* in the sample was carried out using Matrix-Assisted Laser Desorption/Ionization (Maldi-TOF). A subsequent antimicrobial sensitivity test was conducted using the Kirby–Bauer method from a pure inoculum of the microorganism under study with a turbidity corresponding to the suspension of the MC Farland 0.5 standard, which is equivalent to 1.5 × 10^8^ cfu/mL, and the inoculum was seeded in the Mueller–Hinton medium and tested for its susceptibility to antibiotics. In the case of 90 mm diameter Mueller–Hinton agar, a maximum of six disks (each 6 mm in diameter) must be placed with 25 mm between them. Incubation was carried out at 37 °C ± 1 °C in an aerobic O_2_ atmosphere for 24 h. In addition to bacterial sensitivity, the laboratory categorizes the antibiotics registered for use in cattle in Spain, to which the bacteria are Sensitive (S), Intermediate (I), or Resistant (R) [12,13], and the category to which the antibiotic belongs for its prudent use on farms, according to the National Antibiotic Resistance Plan. This Plan classifies antibiotics in four groups: Group A, not authorized for veterinary use; Group D, cataloged as first choice; Group C, cataloged for precautionary use; Group B, cataloged for restricted use, as they can only be used if there are no Group D or C antimicrobials that can be used in clinical treatment [14].

## 3. Results

Platelet culture showed *H. somni* as a unique pathogen isolated from the sample of the uterine exudate submitted to microbiological analysis (Figure 2). After the Kirby–Bauer disc diffusion method, the study showed that *H. somni* was sensitive to Amoxicilin-Clavulamic Acid (Group D); Tiamulin, Florfenicol, and Tilmicosin (Group C); and Enrofloxacin and Marbofloxacin (Group B) (see Table 1 for detailed data).

The cow was under daily clinical examination by the veterinarian and evolved towards healing from the first day of treatment. The cow was considered fully recovered on day 9 after the onset of PM diagnosis (21 DIM) when it no longer showed any clinical signs of systemic disease, reached its expected milk production (>30 L/day), no uterine discharge was observed, and the ultrasound images of all parts of the uterus were completely healthy and showed almost a complete uterine involution.

## 4. Discussion

PM is a prevalent uterine disease in dairy cows during the first 21 days postpartum which can directly compromise the cow’s life [3]. It is assumed that PM is a polymicrobial-caused disease. A wide range of pathogens are described to be causative of metritis in dairy cows, such as *Bacteroides* and Fusobacterium [15]. However, a wide range of other potential pathogens such as *H. somni* have occasionally been identified in metritic cows with an uncertain role in the pathogenesis of uterine diseases [16]. To the authors’ knowledge, this is the first time that *H. somni* was isolated as the sole pathogen in a case of PM.

Although culture-based diagnosis may have limitations compared to other techniques for pathogen identification, it is accepted and widely used [17], above all at an on-farm level during routine and feasible diagnosis. Similarly, the Kirby–Bauer disc diffusion method, although it has limitations compared to other techniques related to antimicrobial resistance and MIC determination, is also a reliable and feasible method to make on-farm decisions about the choice of the use of the best antibiotics to cure animal diseases, but also maximizing the One Health principle. Since we did not find prior results of *H. somni* as a single pathogen in positive bovine metritis, and we also did not have an interpretation in the main databases (Clinical and Laboratory Standards Institute -CLSI- and European Committee on Antimicrobial Susceptibility Testing -EUCAST-) for bovine reproductive diseases, we decided to use the Kirby–Bauer technique, and obtained S, I, and R results as an initial study.

Unlike other studies that have investigated the susceptibility of *H. somni* to antimicrobials [18], this clinical case reports the identification of a pathogen resistant to tetracyclines and cephalosporins (Ceftiofur), antibiotics commonly used to treat uterine diseases in dairy cows. This points out the importance of investigating clinical cases routinely to decide the best antimicrobial treatment and to avoid the use of repeating antibiotic treatments not supported by laboratory analysis, which can increase multi-drug resistance in *H. somni* and other bacterial agents [19].

Previous studies have described the consistent presence of *H. somni* in the uteruses of healthy cows or cows suffering from uterine pathologies such as vaginitis, cervicitis, metritis, or endometritis [7,15,16]. In a study carried out in South Africa, it seems that the authors also observed the possibility that *H. somni* could act as a primary pathogen in these cases [20]. However, we could not find previous studies that report the isolation of *H. somni* as the only pathogen present in PM. As mentioned above, *H. somni* was never isolated in any vaginal or uterine content on this farm before. Furthermore, it is quite uncommon to only isolate one bacteria from uterine samples. Fortunately, the laboratory that carries out such routine analyses for this farm always keeps the samples for a period as a “safety method” in case there is a doubt on the results or procedure. In this case, due to the uniqueness of the result (isolation of *H. somni* alone, without any other bacteria), the culture was repeated by the laboratory following the same methods, with the same results confirming the findings.

This report describes for the first time a case of PM caused by *H. somni* as a single primary pathogen. This fact should lead researchers to study as many cases of metritis in cows as possible, including *H. somni* in laboratory diagnoses on a routine basis, in order to know the real extent of the incidence of this microorganism on uterine diseases in cattle. In the event that the incidence of *H. somni* is remarkably high or frequent, further research into the use of commercial vaccines destined for the prevention of BRD in cattle, which could perhaps be useful for the prevention of reproductive pathology caused by *H. somni* in cows, should be considered.

## 5. Conclusions

This report highlights the first description of PM in a dairy cow caused by *H. somni* as a unique isolated agent in culture-based diagnosis. Veterinarians, practitioners, and farmers should consider this pathogen a causative agent of uterine disease and include it in laboratory diagnosis for both isolation and antibiotic-resistance studies, especially to increase the efficacy of antimicrobial treatments and maximize the One Health principle at the farm level.

Further studies are needed to specify the importance of *H. somni* as a causative agent of PM in dairy cows and to elucidate whether the use of commercial vaccines against BRD could be useful to prevent reproductive pathology caused by *H. somni* in cows.

## Figures and Tables

**Figure 1 vetsci-11-00117-f001:**
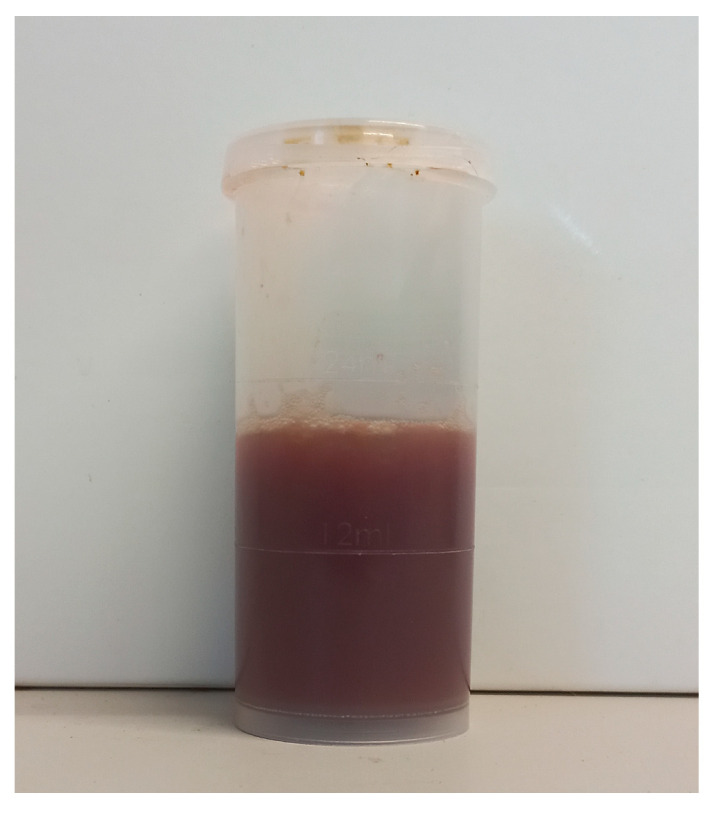
Collected piohemorrhagic, watery red-brown uterine exudate submitted to the laboratory.

**Figure 2 vetsci-11-00117-f002:**
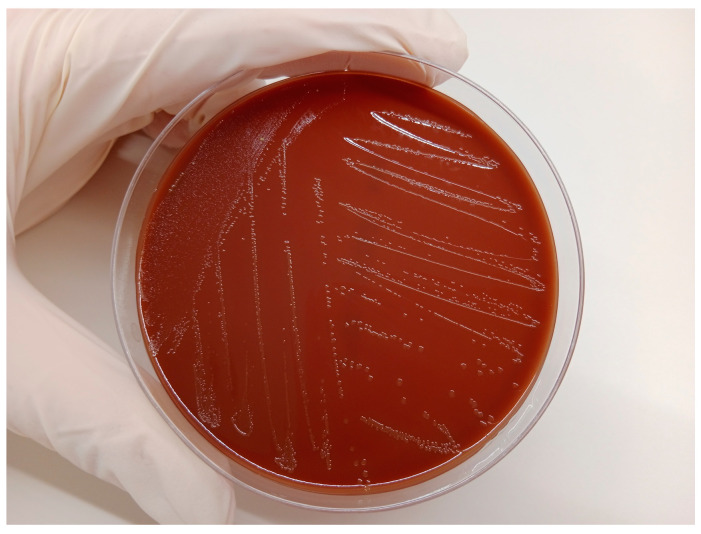
Plate culture in a chocolate agar media (Biomérieux Ref. 43101, Marcy-l’Etoile, France) positive for *Histophillus somni*.

**Table 1 vetsci-11-00117-t001:** Results of the sensitivity of *Histophilus somni* to antibiotics through the Kirby–Bauer method.

Group ^1^	Antibiotic Family	Antibiotic	Disc Concentration	Result for *Histophilus somni* ^2^
D	Beta-lactams	Penicillin G	6 µg	R
D	Beta-lactams	Ampicillin	10 µg	R
D	Beta-lactams	Amoxicillin-Clavulanic Acid	30 µg	S
D	Tetracyclines	Tetracycline	30 µg	R
D	Tetracyclines	Doxycycline	30 µg	I
D	Sulfonamides	Sulfamide-Trimethroprim	75 µg	R
D	Aminocyclitol	Spectinomycin	100 µg	R
C	Pleuromutilines	Tiamulin	30 µg	S
C	Anfenicols	Florfenicol	30 µg	S
C	Macrolides	Tilmicosin	15 µg	S
C	Macrolides	Tylosin	10 µg	I
C	Aminoglycosides	Gentamicin	30 µg	R
C	Aminoglycosides	Neomicin	30 µg	R
C	Aminoglycosides	Apramycin	15 µg	R
B	Fluoroquinolones	Enrofloxacin	5 µg	S
B	Fluoroquinolones	Marbofloxacin	5 µg	S
B	Polymixins	Colistin	50 µg	I
B	Cefalosporines	Ceftiofur	30 µg	R
B	Cefalosporines	Cefquinoma	30 µg	I

^1^ According to the program of prudent use of antimicrobials and the prescription guidelines associated with the National Antibiotic Resistance Plan, these are classified into 4 groups: Group A, not authorized for veterinary use; Group D, cataloged as first choice; Group C, cataloged for precautionary use, Group B, cataloged for restricted use, as they can only be used if there are no Group D or C antimicrobials that can be used in clinical treatment. ^2^ Sensitive (S); Intermediate (I); Resistant (R) Analytical Method. PNT/P21-001. Kirby–Bauer.

## Data Availability

Data are contained within the article and Appendix A.

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
