# Peer review of "Histophilus somni* as a Unique Causative Agent of Puerperal Metritis (PM) in a Third-Lactation Holstein Cow"

_vetsci, 2024, doi:10.3390/vetsci11030117_

Round 1
Reviewer 1 Report (Previous Reviewer 1)
Comments and Suggestions for Authors
The expanding information in the articule improve de quality of the paper
Author Response
Dear Reviewer,
Thanks for your positive comments and report, and the time spent in reviewing thie manuscript.
Reviewer 2 Report (Previous Reviewer 2)
Comments and Suggestions for Authors
In my opinion the changes and additions made adequately complete this case report
Author Response
Dear Reviewer,
Thanks for your positive comments and report, and the time spent in reviewing thie manuscript.
Reviewer 3 Report (Previous Reviewer 3)
Comments and Suggestions for Authors
I recommend accept in present form.
Author Response
Dear Reviewer,
Thanks for your positive comments and report, and the time spent in reviewing thie manuscript.
Reviewer 4 Report (Previous Reviewer 4)
Comments and Suggestions for Authors
The manuscript has notably improved since its initial version as the authors have effectively addressed the majority of the reviewers' comments.
Author Response
Dear Reviewer,
Thanks for your positive comments and report, and the time spent in reviewing thie manuscript.
This manuscript is a resubmission of an earlier submission. The following is a list of the peer review reports and author responses from that submission.
Round 1
Reviewer 1 Report
Comments and Suggestions for Authors
The clinical examen is not complete, lack dates about ketosis and subclinical hipocalcemia.
The result of cultivation with only a bacteria is very strrange
does not speak about other problens in the farm related with de bacteria like brd or neuroñlogical problem in calves

Reviewer 2 Report
Comments and Suggestions for Authors
The work reports an interesting clinical case carefully described. The authors describe for the first time a case of metritis associated exclusively with H. somnum.
The description of the case would be more complete with a brief overview of the virulence factors of H. somnus, the pathogenesis of the host and the immune response to this bacterium in order to better define what is indicated in the conclusions and maximize the One Health principle.
The bibliography should be integrated.
Reviewer 3 Report
Comments and Suggestions for Authors
Title of the Manuscript: “Histophilus somni as a Unique Causative Agent of Puerperal Metritis (PM) in a Third Lactation Holstein Cow” by Molín et al.
Introduction:
Suggested Revision: Replace “after a cow gives birth” with “after calving” for specificity.
Additional Information Request: Elaborate on the incidence of metritis in cows, preferably in percentage terms. Also, discuss primary diseases associated with metritis such as milk fever, retained placenta, etc. This section requires more detailed information.
Case Presentation:
Lines 65-66: The statement “Technical management of the animals, farm biosecurity, and welfare conditions were excellent” needs substantiation. Please provide criteria or methods used for this assessment.
Lines 72-73: Regarding the statement “Cows were fed a total mixed ration consisting of corn silage, grass silage, and concentrate,” it would be beneficial to specify the composition of the Total Mixed Ration (TMR) in percentages, as well as its chemical composition, including Net Energy for Lactation (NEL), Crude Protein (CP), Acid Detergent Fiber (ADF), Neutral Detergent Fiber (NDF), etc.
Line 73: In the context of “Cows were milked three times per day,” please provide more details about the milking system used on this farm.
Line 83: Specify the software mentioned in “...the software alerted.” Include the name and the company of the software.
Line 85: Clarify if the clinical examination was performed by a local veterinarian and provide additional details.
Line 88: For the statement “β-hydroxybutyrate (BHB) concentration in blood ≥0.5 mmol/L,” indicate the equipment used for determining BHB levels.
Line 94: Regarding “oral calcium (Bovicalk, Boehringer Ingelheim, Germany) every 24 hours for 2 days,” clarify how hypocalcemia was identified, or if this was an additional treatment.
Lines 132-136: The description of the cow's recovery needs more detail. Please specify how frequently clinical examinations were performed and what the expected milk production was.
Results:
Please provide information on the number of cows examined during this study and identify other pathogens that were found.
This revised text enhances clarity, asks for specific details where needed, and maintains the original intent of the review.
Reviewer 4 Report
Comments and Suggestions for Authors
This case report points to the occurrence of postpartum metritis apparently caused only by Histophilus somni, which, to my knowledge, has not yet been reported, even though this agent has already been isolated in infections of the genitourinary tract of cattle.
The laboratory methodology that the authors used seems sufficiently convincing to me and the results suggest that more use should be made of bacterial culture, together with antibiotic susceptibility tests, especially in cases of postpartum uterine infection with low therapeutic success. Most field clinicians almost always use antibiotics aimed at the most frequently isolated agents such as Trueperella pyogenes and Fusobacterium necrophorum, and even between these agents there may be significant differences in therapeutic response.
However, I also recognize potential areas for improvement in this manuscript. The subsequent points highlight certain weaknesses that, in my opinion, deserve consideration.
Lines 65-67, the authors refer that the “Technical management of the animals, farm biosecurity and welfare conditions were excellent.” In scientific manuscripts, the incorporation of subjective opinions is generally unconventional. It may be more advisable to employ a comparative term such as "above average" substantiated by appropriate justification, to enhance precision in communication.
Lines, 79-82, the authors refer that “Quarterly, samples of uterine contents were taken from cows suffering from uterine disease through uterine lavage and subjected to laboratory diagnosis to identify the pathogens involved and monitor antimicrobial sensibilities.” Enumerating the most frequently isolated bacterial agents in this farm would perhaps enrich the manuscript.
Lines 94-102, the author refer that an “antimicrobial treatment of 15 mg/kg of BW of amoxycillin trihidrate every 48h by intramuscular route three times” was included in the usual treatment protocol for PM on this dairy farm. Considering that this antibiotic is not one of the most frequently used for these clinical conditions, how do you justify the choice for inclusion in the treatment protocol for cows with PM? Based on results of previous sensibility tests in PM cases in that farm?
Line 153, replace ”cancan” with “can”.
Line 178, I suggest “H. somni and other bacterial agents”.
One of the most comprehensive reviews on Histophilus somni and reproductive disease in the cow is not included in the references:
- Kwiecien, J. M., & Little, P. B. (1991). Haemophilus somnus and reproductive disease in the cow: A review. The Canadian Veterinary Journal, 32(10), 595.
As the authors already know, before 2003, it was believed that Haemophilus somnus, Histophilus ovis and Histophilus agni were distinct species. However, they are now collectively classified as Histophilus somni.
Comments on the Quality of English LanguageLine 57, replace “cowcaused” with “cow caused”.
Lines 88-90, “This case of PM was taken advantage of to obtain uterine samples for the routine laboratory analysis.” Kindly revise this sentence as its intended meaning is not distinctly clear.
Line 153, replace ”cancan” with “can”.
Reviewer 5 Report
Comments and Suggestions for Authors
Thank you for selecting me as a reviewer of this manuscript.
Please confirm my comments to the authors.
I am sorry for delaying sending my comments.